

# Constructing three emotion knowledge tests from the invariant measurement approach

Ana R. Delgado[1], Gerardo Prieto[1] and Debora I. Burin[2]

[1] Faculty of Psychology, Universidad de Salamanca, Salamanca, Spain
[2] Faculty of Psychology, Universidad de Buenos Aires-CONICET, Buenos Aires, Argentina

## ABSTRACT

**Background**. Psychological constructionist models like the Conceptual Act Theory (CAT) postulate that complex states such as emotions are composed of basic psychological ingredients that are more clearly respected by the brain than basic emotions. The objective of this study was the construction and initial validation of Emotion Knowledge measures from the CAT frame by means of an invariant measurement approach, the Rasch Model (RM). Psychological distance theory was used to inform item generation.
**Methods**. Three EK tests—emotion vocabulary (EV), close emotional situations (CES) and far emotional situations (FES)—were constructed and tested with the RM in a community sample of 100 females and 100 males (age range: 18–65), both separately and conjointly.
**Results**. It was corroborated that data-RM fit was sufficient. Then, the effect of type of test and emotion on Rasch-modelled item difficulty was tested. Significant effects of emotion on EK item difficulty were found, but the only statistically significant difference was that between "happiness" and the remaining emotions; neither type of test, nor interaction effects on EK item difficulty were statistically significant. The testing of gender differences was carried out after corroborating that differential item functioning (DIF) would not be a plausible alternative hypothesis for the results. No statistically significant sex-related differences were found out in EV, CES, FES, or total EK. However, the sign of $d$ indicate that female participants were consistently better than male ones, a result that will be of interest for future meta-analyses.
**Discussion**. The three EK tests are ready to be used as components of a higher-level measurement process.

# INTRODUCTION

Contrary to the view of emotions as discrete natural events, an amalgam of expression and behavior with a distinct neural basis, constructionism posits that they are not "ontologically objective" categories or brute facts (*Barrett, 2012*; *Searle, 2010*) but ontologically subjective categories. These categories depend on collective intentionality, and not only physical actions and body states. That some of the so-called emotional behaviors (e.g., fight or freeze) have innate circuits does not imply that discrete emotions have them too

Corresponding author
Ana R. Delgado, adelgado@usal.es

(*Lindquist et al., 2012*). At the neural level, there are no bi-univocal correspondences between a given emotion and areas of activation (*Lindquist et al., 2012*), and a variety of emotional experiences are associated with dynamic interactions of extended neural networks (*Raz et al., 2016*).

If we consider emotion categories as ontologically subjective categories, then we can think of them as cognitive tools allowing us to represent the shared meaning of changes in the natural world, i.e., the shared meaning of both internal physical changes and of sensory changes external to the perceiver (*Barrett, 2012*). Psychological constructionist models such as the Conceptual Act Theory (CAT) postulate that complex states (e.g., emotions and cognitions) are composed of basic psychological ingredients that are brain-based (*Barrett, 2009a*). The CAT hypothesizes that physical changes are transformed into emotions when taking on psychological functions that require socially shared conceptual knowledge to be meaningful to the perceiver; it is in this sense that emotions are real: they are both biologically evident and part of our social reality (*Barrett, 2006*; *Barrett, 2012*; *Barrett, 2014*; *Wilson-Mendenhall et al., 2011*). Some emotion categories serve this purpose only for members of one particular culture, but there are some others, e.g., happiness, sadness, anger, fear, and disgust, that can be thought of as closer to universal and so it is typical to find them in experimental and developmental studies (*Lindquist et al., 2014*; *Tracy & Randles, 2011*).

In any case, emotion categories are not context-independent representations: Emotion knowledge (EK) is situated (*Barrett, 2012*; *Barrett, 2014*; *Wilson-Mendenhall et al., 2011*), and thus there are cultural and individual differences in the use of emotion words, a skill that is closely related to emotional intelligence (*Barrett, 2009b*). Currently, it is not clear whether ability-based emotional intelligence is a construct with the same status as fluid and crystallized intelligence or rather whether it is already defined by extant constructs, such as acculturated knowledge/crystallized intelligence (*MacCann et al., 2014*). The predominant operationalization has been the Mayer-Salovey-Caruso Emotional Intelligence Test Battery, whose psychometrical properties are not optimal (*Orchard et al., 2009*). Given that emotional aptitude variables predict dependent variables as relevant as perceived stress (*Rey, Extremera & Pena, 2016*) or depressive symptoms (*Luque-Reca, Augusto-Landa & Pulido-Martos, 2016*) we should start to test narrowly defined emotion domains each requiring its own theories and measures (*Matthews, Zeidner & Roberts, 2012*).

The fact that both categorical knowledge and contextual information are *constitutive* of emotions is a substantive reason to prefer invariant measurement models over the reflective structural equation models usually employed in validation studies; there are also psychometric reasons (e.g., *Engelhard & Wang, 2014*) to prefer invariant measurement models to the formative structural equation models recommended by *Coan & Gonzalez (2015)* in the CAT context. An implementation of the invariant measurement approach is the Rasch model (RM; *Rasch, 1960*), increasingly used to validate psychological and neuropsychological tests (*Delgado, 2012*; *Engelhard & Wang, 2014*; *Miguel, Silva & Prieto, 2013*; *Prieto et al., 2010*). The probability that subject $n$ passes item $i$ is modeled as $Pni = \exp(Bn - Di)/(1 + \exp[Bn - Di])$, where $Bn$ is the person level and $Di$ is the item location. This logistic one-parameter model shows the property of *specific objectivity*, allowing the
algebraic separation of items and person parameters (the person parameter can be erased when estimating the item parameters). This is so because the sum score for an item or person is a *sufficient* statistic for the corresponding parameter, i.e., it captures all the information about the corresponding parameter that is contained in the sample.

One of the main advantages of the RM derives from the fact that it is a conjoint measurement model: If empirical data fit the model adequately, then person measures (e.g., aptitude, personality trait) and item locations (e.g., difficulty, severity) can be jointly located on an *interval* scale (variable map) whose unit is the *logit*. When using the RM, item parameter estimations are sample-independent and person parameter estimations are independent of the particular items that have been used; this is not true of the classical measurement model. An item of great interest for psychological measurement is the fact that the level of analysis is the individual in the RM, while Structural Equation Models (which are statistical models for covariances) use the group as level of analysis (*Engelhard & Wang, 2014*).

Thus, the general objective of this study was the construction and initial validation of EK measures from a psychological constructionist theoretical frame, the CAT, by means of an invariant measurement approach, the RM. When validating situational tests of emotion understanding, it has been found that items describing situations with close/concrete receivers are easier than those in which receivers are far/abstract (*Delgado, 2016*), a result that is predicted by *psychological distance* theories (*Soderberg et al., 2015*; *Trope & Liberman, 2010*), and so the close/far distinction has been taken into account in the generation of items for the situational tests. Three EK tests have been constructed and tested with the RM, both separately—vocabulary, close and far situations—and conjointly, given that they are all EK measures.

## MATERIALS & METHODS

### Participants
The sample was composed of 100 females and 100 males, with ages ranging from 18 to 65 years old, Spanish as first language, and Spanish nationality. Roughly half of them ($n = 94$) were young adults (18–30). The educational level was high (155 participants were or had been to college or further).

Even though the property of specific objectivity allows the person-independent estimation of item parameters, i.e., no representative sample is needed, we obtained the most heterogeneous sample that was available to us by recruiting participants from various Spanish regions in an art museum that was public and open to all.

### Instruments
Three tests were constructed with *LiveCode Ltd. (2011)* and implemented on a portable computer. Identification, gender, age, informed consent, response option and right/wrong answers were asked for and automatically stored by the application. Each of the three tests was composed of 40 multiple-choice items, eight for each of the five emotion "families" of happiness, sadness, anger, fear, and disgust. There was no time limit, and feedback on the total score (number of correct answers; possible range: 0–120) was provided in the last

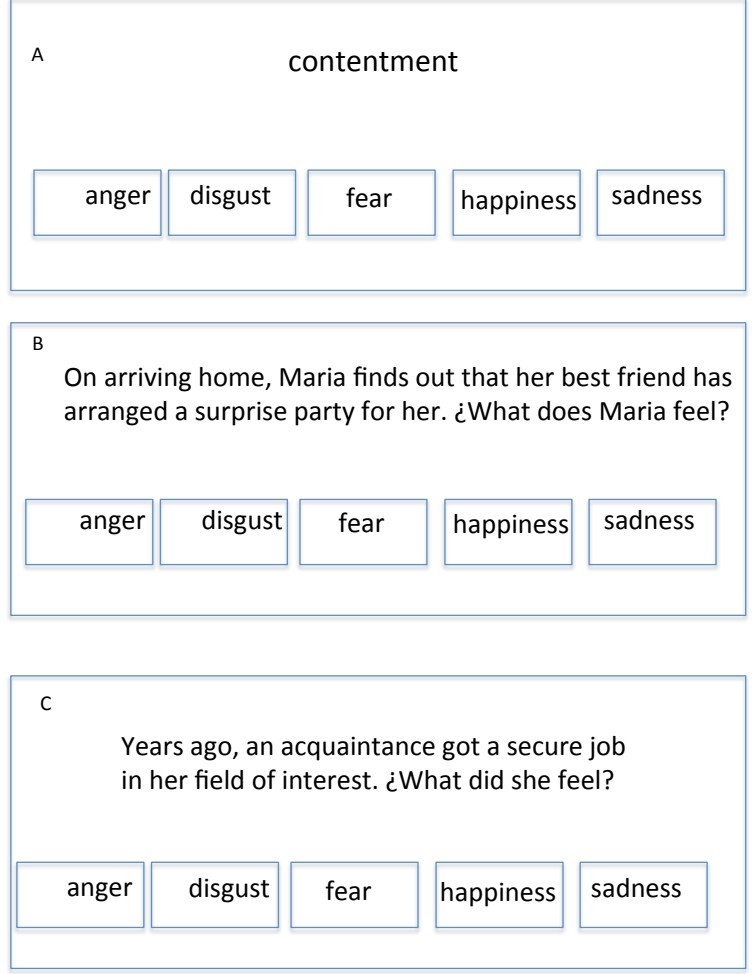

**Figure 1** **Three HAPPINESS item examples: (A) EV item. (B) CES item. (C) FES item.** Note: Items were written in Spanish, so the translation is an approximation.

screen. Tests are described below in the order they were applied. Item examples can be seen in Fig. 1.

### Emotion Vocabulary (EV)

Each item stem was an emotion word, carefully selected from the corpus of the Royal Spanish Academy CORPES XXI, which has 25 million of forms for each year between 2001 and 2012 (*Real Academia Española, 2015*). The five response options, of which only one is correct, were happiness, sadness, anger, fear, and disgust (*alegría, tristeza, ira, miedo,* and *asco,* in standard Spanish). The subject had to choose the response option whose meaning was the closest to that of the target word.

### Close Emotional Situations (CES)

Item stems were verbal scenarios showing a character and a close/concrete moment, act, object and place. Scenarios described concrete variations of the prototypes of the five

emotion "families". Some 40 first names (half of them male) were selected from the database of the National Statistical Institute so that each scenario showed a different character identified by his/her name. There were five response options: happiness, sadness, anger, fear, and disgust. This is the adequate level of specificity for this kind of test, given that it has been found that emotional inferences from verbal scenarios are more specific than valence and class, but not more specific than emotion "families" (*Molinari et al., 2009*). The subject had to choose the option that best described the emotion that would be typical to feel in that concrete situation.

### Far Emotional Situations (FES)
Item stems were verbal scenarios showing a far/abstract time, character and situation. Scenarios described abstract variations of the prototypes of the five emotion "families" and the main character was not identified by his/her first name but by a generic label (half of them female, e.g., "a girl"). Response options were the same as in the previous tests, and the task was to choose the one that best described the emotion that would usually be felt in that abstract situation.

## Procedure
Participants were approached by a university researcher with a visible identification card, and asked about age, provenance and first language to warrant inclusion criteria. After asking for consent to use the data for research purposes, the tests were individually applied on a portable computer.

## Data analysis
Responses to the three tests were separately analyzed with the RM. Then, after conjoint scaling of items and persons, the effect of type of test and emotion on item difficulty was probed by means of factorial ANOVA.

Rasch analyses were performed with the computer program Winsteps 3.80.1 (*Linacre, 2013*). Data-model fit is assessed by *outfit* (calculated by adding the standardized square of residuals after fitting the model over items or subjects to form chi-square-distributed variables) and *infit* (an information-weighted form of *outfit*). *Infit/outfit* values over 2 distort the measurement system (*Linacre, 2013*). Unidimensionality is a requirement for the model, not implying that performance is due to a unique psychological process (*Reckase, 1979*); component analyses of residuals are performed by Winsteps 3.80.1 in order to test this assumption. The indications are that Rasch measures should account for at least 20% of the total variance and it is recommended that the unexplained variance in the first contrast be lower than 3 (*Miguel, Silva & Prieto, 2013*).

Differential item functioning (DIF) analysis tests the generalized validity of the measures for different groups. In this case, given that there is evidence of female superiority in the accuracy of affective judgements (*Hall, Gunnery & Horgan, 2016*), a plausible alternative hypothesis is the instrumental one: items could be functioning differently for males and females. Thus, the DIF hypothesis was tested: The standardized difference between item calibrations in the case of two groups (i.e., male and female) was calculated and tested using Bonferroni-corrected *alpha* levels; the Rasch aptitude estimates from the analysis of

all the data were held constant, providing the conjoint measurement scale in *logit* units (*Linacre, 2013*).

## RESULTS

### EV test

One person and two items got extreme scores (i.e., zero or perfect raw scores) and so their Rasch measures were not estimated. The Rasch analysis of the remaining data indicates good data-model fit for items, *mean infit* was .99 ($SD = .08$) and *mean outfit* was .91 ($SD = .31$). For persons, mean *infit* was 1.00 ($SD = .25$) and *mean outfit* was .91 ($SD = .64$). No item showed *infit/outfit* over 1.5. Twelve persons (6%) showed *outfit* over 2, but just one of them showed *infit* over 2. The percentage of variance explained by EV measures was 33.9% and the component analysis of residuals showed that the unexplained variance in the first contrast was 2.3. Finally, *item reliability* (.96) and *model person reliability* (.72) were good enough. Table 1 shows the main results of the item analysis.

Average person aptitude in *logit* units was 2.19, $SD = 1.04$, range $= -1.12$ to 4.78. Just one item (EV24, happiness) showed sex-related DIF favoring male subjects, i.e., male subjects had a higher probability of passing this item than female subjects with the same total score. No gender differences (impact) in Rasch measures were found, *Welch-t* (196) = .68, $p = .50$, $d = -.11$ (conventionally, 0 = female, 1 = male). As an illustration, Table 2 shows the map of the variable, where the right side shows item locations while person measures are situated at the left.

### CES test

Seven persons obtained extreme scores; their Rasch measures were not estimated. The Rasch analysis indicates good data-model fit: Item *mean infit* = .98 ($SD = .11$), *mean outfit* = .87 ($SD = .25$); person mean *infit* = 1.00 ($SD = .17$), *mean outfit* = .87 ($SD = .74$). No item showed *infit/outfit* over 2. Eight persons (4%) showed *outfit* over 2. The percentage of variance explained by close emotional situations measures was 22.9% and the component analysis of residuals showed that the unexplained variance in the first contrast was 2.3. As to *item reliability* and *model person reliability*, they were .93 and .58, respectively. Table 3 shows the main results of the item analysis.

Average person aptitude in *logit* units was 2.66, $SD = .93$, range $= -1.68$ to 4.45. Neither sex-related DIF nor gender significant differences were found, *Welch-t* (182) = 1.39, $p = .17$, $d = -.21$.

### FES test

Four persons got extreme scores. The Rasch analysis of the remaining data indicates good data-model fit: For items, *mean infit* was .97 ($SD = .11$) and *mean outfit* was .88 ($SD = .32$). For persons, mean *infit* was 1.00 ($SD = .18$) and *mean outfit* was .88 ($SD = .59$). No item showed *infit/oufit* over 2; eight persons showed *outfit* over 2. The percentage of variance explained by measures was 22.9% and the component analysis of residuals showed that the unexplained variance in the first contrast was 3.2. Finally, *item reliability* (.93) and *model person reliability* (.65) were acceptable. Table 4 shows the main results of the item analysis.

**Table 1** Emotion vocabulary items: Rasch analysis results.

| Item | Score | Rasch *Di* | *SE* | *Infit* | *Outfit* |
|------|-------|-----------|------|---------|----------|
| 01 | 198 | −2.95 | .72 | 1.01 | 1.13 |
| 02 | 137 | 1.26 | .17 | .94 | .90 |
| 03 | 185 | −.75 | .28 | .88 | .61 |
| 04 | 194 | −1.79 | .43 | .93 | .35 |
| 05 | 194 | −1.79 | .43 | .91 | .86 |
| 06 | 158 | .60 | .19 | .99 | 1.01 |
| 07 | 199 | −3.66 | 1.01 | .98 | .20 |
| 08 | 190 | −1.22 | .34 | .92 | .53 |
| 09 | 162 | .46 | .20 | 1.05 | .99 |
| 10 | 148 | .94 | .18 | 1.04 | 1.00 |
| 11 | 193 | −1.62 | .40 | .82 | .32 |
| 12 | 200 | – | – | – | – |
| 13 | 193 | −1.62 | .40 | .95 | .44 |
| 14 | 189 | −1.11 | 32 | .98 | .62 |
| 15 | 164 | .38 | .20 | 1.04 | .99 |
| 16 | 155 | .71 | .18 | .98 | .88 |
| 17 | 173 | −.02 | .22 | 1.02 | .97 |
| 18 | 200 | – | – | – | – |
| 19 | 165 | .34 | .20 | 1.05 | 1.19 |
| 20 | 186 | −.83 | .29 | 1.03 | .73 |
| 21 | 75 | 2.84 | .16 | 1.06 | 1.27 |
| 22 | 147 | .97 | .18 | .91 | .97 |
| 23 | 95 | 2.34 | .16 | .96 | .99 |
| 24 | 133 | 1.37 | .17 | 1.04 | .96 |
| 25 | 171 | .08 | .22 | 1.13 | 1.22 |
| 26 | 157 | .64 | .19 | .92 | .89 |
| 27 | 192 | −1.47 | .37 | .87 | .37 |
| 28 | 171 | .08 | .22 | .89 | .74 |
| 29 | 172 | .03 | .22 | 1.01 | .87 |
| 30 | 182 | −.53 | .26 | 1.11 | 1.50 |
| 31 | 111 | 1.94 | .16 | 1.23 | 1.43 |
| 32 | 196 | −2.23 | .52 | 1.04 | 1.07 |
| 33 | 170 | .13 | .21 | 1.01 | .87 |
| 34 | 52 | 3.49 | .18 | 1.01 | 1.03 |
| 35 | 177 | −.22 | .24 | .96 | .91 |
| 36 | 175 | −.12 | .23 | .92 | 1.19 |
| 37 | 176 | −.17 | .23 | 1.02 | 1.10 |
| 38 | 190 | −1.22 | .34 | 1.04 | .96 |
| 39 | 53 | 3.46 | .18 | 1.02 | 1.25 |
| 40 | 137 | 1.26 | .17 | 1.00 | .95 |

**Table 2   Emotion vocabulary- variable map.**

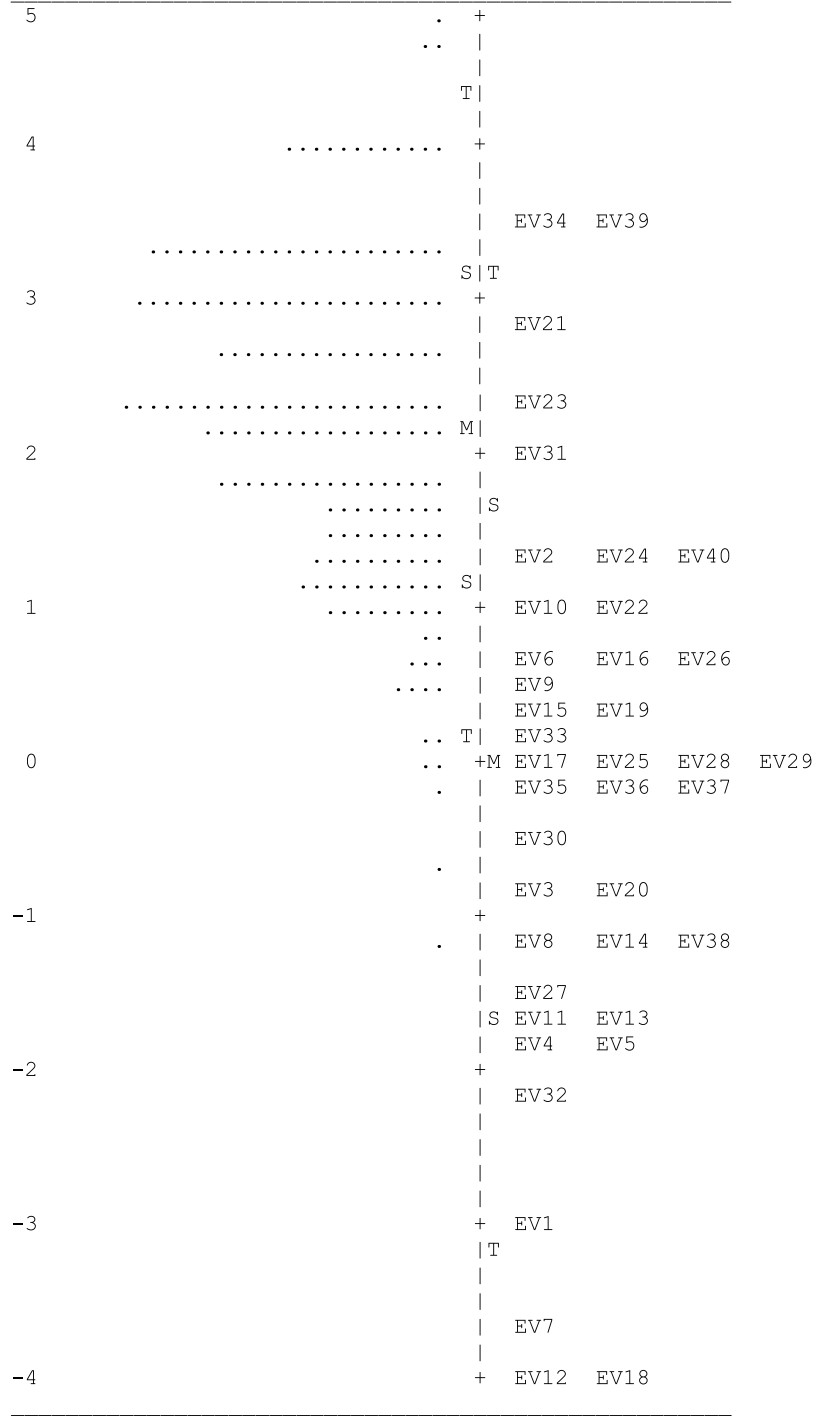

**Notes.**
M, mean; S=1 *SD*; T=2 *SD*.

**Table 3  Close emotional situations items: Rasch analysis results.**

| Item | Score | Rasch $D_i$ | SE | Infit | Outfit |
|------|-------|-------------|-----|-------|--------|
| 01 | 148 | 1.51 | .17 | 1.16 | 1.32 |
| 02 | 190 | −.65 | .34 | 1.06 | .74 |
| 03 | 197 | −2.01 | .61 | 1.07 | .64 |
| 04 | 169 | .76 | .21 | 1.04 | 1.09 |
| 05 | 197 | −2.01 | .61 | .80 | .37 |
| 06 | 123 | 2.19 | .16 | 1.13 | 1.22 |
| 07 | 138 | 1.80 | .17 | 1.08 | 1.03 |
| 08 | 107 | 2.59 | .16 | 1.07 | 1.06 |
| 09 | 197 | −2.01 | .61 | .87 | .83 |
| 10 | 198 | −2.45 | .73 | .80 | .80 |
| 11 | 183 | −.01 | .27 | .94 | .73 |
| 12 | 188 | −.43 | .31 | .89 | .57 |
| 13 | 164 | .96 | .20 | 1.07 | .96 |
| 14 | 160 | 1.11 | .19 | .97 | .96 |
| 15 | 187 | −.34 | .30 | 1.02 | .87 |
| 16 | 139 | 1.77 | .17 | .92 | .89− |
| 17 | 195 | −1.44 | .47 | .88 | .66 |
| 18 | 157 | 1.22 | .19 | .90 | .81 |
| 19 | 161 | 1.08 | .19 | 1.16 | 1.66 |
| 20 | 179 | .25 | .25 | 1.03 | .77 |
| 21 | 182 | .06 | .26 | 1.06 | .98 |
| 22 | 196 | −1.69 | .53 | .74 | .44 |
| 23 | 180 | .19 | .25 | .97 | .68 |
| 24 | 176 | .42 | .23 | 1.04 | 1.00 |
| 25 | 189 | −.54 | .33 | .97 | .83 |
| 26 | 191 | −.77 | .36 | 1.17 | .98 |
| 27 | 195 | −1.44 | .47 | 1.15 | .73 |
| 28 | 187 | −.34 | .30 | .92 | .62 |
| 29 | 185 | −.17 | .28 | .89 | .56 |
| 30 | 166 | .88 | .20 | .97 | .82 |
| 31 | 190 | −.65 | .34 | .99 | 1.13 |
| 32 | 198 | −2.45 | .73 | .81 | 1.07 |
| 33 | 189 | −.54 | .33 | .87 | .61 |
| 34 | 191 | −.77 | .36 | 1.01 | .89 |
| 35 | 192 | −.91 | .38 | .86 | .76 |
| 36 | 173 | .57 | .22 | 1.06 | 1.11 |
| 37 | 121 | 2.24 | .16 | .99 | .94 |
| 38 | 159 | 1.15 | .19 | 1.09 | 1.19 |
| 39 | 194 | −1.24 | .44 | .90 | .55 |
| 40 | 126 | 2.11 | .16 | .99 | .99 |

**Table 4  Far emotional situations items: Rasch analysis results.**

| Item | Score | Rasch $D_i$ | SE | Infit | Outfit |
|------|-------|-------------|-----|-------|--------|
| 01 | 184 | −.35 | .28 | 1.19 | .95 |
| 02 | 160 | .87 | .19 | .97 | .89 |
| 03 | 178 | .06 | .24 | 1.14 | 1.41 |
| 04 | 144 | 1.39 | .17 | 1.09 | 1.25 |
| 05 | 195 | −1.76 | .49 | .79 | .98 |
| 06 | 170 | .47 | .21 | 1.04 | 1.15 |
| 07 | 166 | .64 | .20 | 1.02 | 1.01 |
| 08 | 171 | .42 | .22 | 1.01 | .94 |
| 09 | 161 | .83 | .19 | 1.00 | .94 |
| 10 | 198 | −2.83 | .75 | .79 | .32 |
| 11 | 177 | .11 | .24 | 1.03 | .90 |
| 12 | 160 | .87 | .19 | .96 | 1.02 |
| 13 | 196 | −2.02 | .54 | .87 | .26 |
| 14 | 190 | −.93 | .35 | .95 | 1.47 |
| 15 | 164 | .72 | .20 | 1.06 | .98 |
| 16 | 176 | .17 | .23 | .98 | .86 |
| 17 | 115 | 2.15 | .16 | .97 | 1.13 |
| 18 | 175 | .22 | .23 | .97 | .78 |
| 19 | 142 | 1.44 | .17 | 1.02 | 1.02 |
| 20 | 191 | −1.05 | .37 | .85 | .46 |
| 21 | 82 | 2.96 | .16 | 1.17 | 1.20 |
| 22 | 171 | .42 | .22 | 1.04 | 1.15 |
| 23 | 184 | −.35 | .28 | .91 | .58 |
| 24 | 158 | .94 | .19 | .97 | 1.14 |
| 25 | 192 | −1.20 | .39 | .83 | .44 |
| 26 | 175 | .22 | .23 | .95 | .68 |
| 27 | 195 | −1.76 | .49 | .71 | .21 |
| 28 | 171 | .42 | .22 | .95 | .81 |
| 29 | 192 | −1.20 | .39 | 1.02 | .81 |
| 30 | 192 | −1.20 | .39 | .85 | .64 |
| 31 | 145 | 1.36 | .17 | 1.09 | 1.06 |
| 32 | 189 | −.81 | .33 | .85 | .52 |
| 33 | 194 | −1.54 | .45 | .82 | .53 |
| 34 | 185 | −.43 | .29 | .97 | .74 |
| 35 | 122 | 1.98 | .16 | 1.24 | 1.69 |
| 36 | 194 | −1.54 | .45 | .81 | .73 |
| 37 | 168 | .55 | .21 | 1.07 | 1.10 |
| 38 | 165 | .68 | .20 | .97 | .92 |
| 39 | 185 | −.43 | .29 | .97 | .88 |
| 40 | 186 | −.51 | .30 | .95 | .56 |

Average person aptitude in *logit* units was 2.46, $SD = 1.09$, *range* $= -2.44$ to 4.34. No item showed sex-related DIF. No gender differences in Rasch measures were found *Welch-t* $(183) = .14$, $p = .89$, $d = -.02$ (conventionally, code was $0 =$ female, $1 =$ male).

## Total EK score

Two items had extreme scores and so their Rasch scores were not estimated. The Rasch analysis of the responses to the remaining 118 items indicates good data-model fit: for items, *mean infit* was .99 ($SD = .07$) and *mean outfit* was .90 ($SD = .25$). For persons, mean *infit* was 1.00 ($SD = .14$) and *mean outfit* was .90 ($SD =. 42$). No item showed *infit/outfit* over 2, and just six persons out of 200 showed *outfit* over 2. The percentage of variance explained by EK measures was 22.7% and the component analysis of residuals showed that the unexplained variance in the first contrast was 4.5 (2.9%). Finally, *item reliability* (.94) and *model person reliability* (.83) were good. Table 5 shows the main results of the item analysis.

Average person aptitude in *logit* units was 2.34, $SD = .75$, *range* = -1.11 to 4.91. No item showed sex-related DIF; significant gender differences (impact) in Rasch measures were not found *Welch-t* $(193) = 1.18$, $p = .24$, $d = -.17$ (conventionally, code was $0 =$ female, $1 =$ male).

In regard to the type of test, item mean difficulty (in *logit* units and in ascending order) was $-0.26$ (CES), $-0.01$ (FES) and 0.28 (EV). As to emotions, item mean difficulty (in ascending order) was $-1.54$ (HAPPINESS), $-0.07$ (FEAR), 0.26 (SADNESS), 0.56 (ANGER), and 0.66 (DISGUST). A factorial ANOVA of the effects of type of test and emotion on EK item difficulty was statistically significant, $F (14, 103) = 5.09$, $p < .001$. Neither type of test, $F(2, 103) = 1.78$, $p = .17$, nor the interaction effects $F(8, 103) = 1.06$, $p = .40$, were statistically significant. Emotion effects on EK item difficulty were found, $F (4, 103) = 14.02$, $p = p < .001$; Bonferroni *post hoc* tests indicated that the only statistically significant difference was that between HAPPINESS and the remaining emotions.

## DISCUSSION

Three emotion knowledge tests have been constructed from a psychological constructionist theoretical frame: one vocabulary and two situational tests. Although items were generated in Spanish, the careful selection of the emotion words and the substantive background (conceptual act and psychological distance theories) should facilitate their adaptation to other languages and/or cultures. Test items and specifications will be made available upon request to accredited researchers for non-commercial purposes.

The RM, an invariant measurement approach, was used for the initial validation of the three tests separately—EV, CES and FES—and conjointly, given that all the items were designed to provide EK measures. In the four cases, data-model fit was good enough, so the probability of a response can be expressed as an additive function of a person parameter and an item parameter; this is consistent with the quantitative assumption implicitly made—but not tested—in most psychological assessment situations (*Michell, 1999*). Even though the first contrast of the component analysis of residuals was slightly over the recommended value for one of the tests, as well as for the conjoint scaling of the three

**Table 5** Emotion knowledge items: Rasch analysis results.

| Item | EMOTION | Score | Rasch *Di* | *SE* | *Infit* | *Outfit* |
|------|---------|-------|-----------|------|---------|----------|
| 001 | HAPPINESS | 198 | −2.59 | .72 | 1.05 | 1.25 |
| 002 | DISGUST | 137 | 1.48 | .16 | .98 | 1.01 |
| 003 | ANGER | 185 | −.41 | .28 | .96 | .70 |
| 004 | FEAR | 194 | −1.43 | .42 | .87 | .53 |
| 005 | DISGUST | 194 | −1.43 | .42 | 1.00 | 1.13 |
| 006 | DISGUST | 158 | .87 | .18 | .98 | .94 |
| 007 | HAPPINESS | 199 | −3.30 | 1.01 | 1.02 | .72 |
| 008 | SADNESS | 190 | −.87 | .33 | .92 | .76 |
| 009 | ANGER | 162 | .74 | .19 | 1.04 | 1.01 |
| 010 | DISGUST | 148 | 1.18 | .17 | .99 | .96 |
| 011 | SADNESS | 193 | −1.27 | .39 | .96 | .44 |
| 012 | HAPPINESS | 200 | – | – | – | – |
| 013 | HAPPINESS | 193 | −1.27 | .39 | 1.00 | .61 |
| 014 | SADNESS | 189 | −.77 | .32 | 1.05 | .94 |
| 015 | DISGUST | 164 | .66 | .19 | 1.00 | .97 |
| 016 | FEAR | 155 | .97 | .18 | .98 | .94 |
| 017 | FEAR | 173 | .29 | .22 | 1.00 | 1.10 |
| 018 | HAPPINESS | 200 | – | – | – | – |
| 019 | ANGER | 165 | .63 | .19 | 1.03 | 1.02 |
| 020 | HAPPINESS | 186 | −.49 | .29 | 1.09 | 1.07 |
| 021 | FEAR | 75 | 2.91 | .15 | 1.12 | 1.14 |
| 022 | ANGER | 147 | 1.21 | .17 | .96 | .88 |
| 023 | DISGUST | 95 | 2.46 | .15 | 1.01 | .98 |
| 024 | HAPPINESS | 133 | 1.58 | .16 | 1.04 | 1.09 |
| 025 | FEAR | 171 | .38 | .21 | 1.06 | 1.14 |
| 026 | SADNESS | 157 | .91 | .18 | .98 | .99 |
| 027 | ANGER | 192 | −1.12 | .37 | 1.00 | .48 |
| 028 | SADNESS | 171 | .38 | .21 | .97 | .86 |
| 029 | HAPPINESS | 172 | .34 | .21 | 1.03 | .95 |
| 030 | DISGUST | 182 | −.20 | .26 | 1.09 | .99 |
| 031 | FEAR | 111 | 2.10 | .15 | 1.18 | 1.30 |
| 032 | FEAR | 196 | −1.87 | .51 | 1.04 | .93 |
| 033 | SADNESS | 170 | .42 | .21 | .99 | 1.02 |
| 034 | DISGUST | 52 | 3.50 | .17 | 1.02 | 1.01 |
| 035 | ANGER | 177 | .09 | .23 | 1.04 | 1.03 |
| 036 | SADNESS | 175 | .20 | .22 | .96 | .85 |
| 037 | SADNESS | 176 | .14 | .23 | 1.04 | 1.08 |
| 038 | FEAR | 190 | −.87 | .33 | 1.03 | .90 |
| 039 | ANGER | 53 | 3.47 | .17 | 1.12 | 1.31 |
| 040 | ANGER | 137 | 1.48 | .16 | 1.03 | 1.01 |
| 041 | ANGER | 148 | 1.18 | .17 | 1.12 | 1.19 |
| 042 | FEAR | 190 | −.87 | .33 | 1.03 | .86 |
**Table 5** (*continued*)

| Item | EMOTION | Score | Rasch *Di* | SE | *Infit* | *Outfit* |
|------|---------|-------|-----------|-----|--------|---------|
| 043 | FEAR | 197 | −2.17 | .59 | 1.03 | 1.00 |
| 044 | FEAR | 169 | .47 | .20 | .99 | .98 |
| 045 | HAPPINESS | 197 | −2.17 | .59 | .85 | .29 |
| 046 | ANGER | 123 | 1.82 | .15 | 1.07 | 1.08 |
| 047 | SADNESS | 138 | 1.45 | .16 | 1.10 | 1.09 |
| 048 | DISGUST | 107 | 2.19 | .15 | 1.10 | 1.10 |
| 049 | HAPPINESS | 197 | −2.17 | .59 | .90 | 1.05 |
| 050 | HAPPINESS | 198 | −2.59 | .72 | .87 | .61 |
| 051 | FEAR | 183 | −.27 | .26 | .96 | .90 |
| 052 | SADNESS | 188 | −.67 | .31 | .89 | .67 |
| 053 | ANGER | 164 | .66 | .19 | 1.03 | 1.00 |
| 054 | ANGER | 160 | .80 | .19 | .99 | .94 |
| 055 | ANGER | 187 | −.58 | .30 | 1.01 | .92 |
| 056 | DISGUST | 139 | 1.43 | .16 | 1.04 | 1.03 |
| 057 | HAPPINESS | 195 | −1.63 | .46 | .87 | .39 |
| 058 | DISGUST | 157 | .91 | .18 | 1.02 | 1.02 |
| 059 | SADNESS | 161 | .77 | .19 | 1.11 | 1.60 |
| 060 | DISGUST | 179 | −.02 | .24 | 1.06 | 1.06 |
| 061 | ANGER | 182 | −.20 | .26 | .97 | .87 |
| 062 | HAPPINESS | 196 | −1.87 | .51 | .82 | .39 |
| 063 | SADNESS | 180 | −.08 | .24 | .97 | .78 |
| 064 | FEAR | 176 | .14 | .23 | .93 | .79 |
| 065 | FEAR | 189 | −.77 | .32 | .95 | .75 |
| 066 | DISGUST | 191 | −.99 | .35 | 1.08 | .93 |
| 067 | FEAR | 195 | −1.63 | .46 | 1.05 | .86 |
| 068 | ANGER | 187 | −.58 | .30 | .96 | .82 |
| 069 | DISGUST | 185 | −.41 | .28 | .93 | .70 |
| 070 | SADNESS | 166 | .59 | .20 | 1.02 | .97 |
| 071 | ANGER | 190 | −.87 | .33 | .99 | 1.37 |
| 072 | HAPPINESS | 198 | −2.59 | .72 | .87 | 1.43 |
| 073 | SADNESS | 189 | −.77 | .32 | .89 | .74 |
| 074 | FEAR | 191 | −.99 | .35 | .98 | 1.05 |
| 075 | HAPPINESS | 192 | −1.12 | .37 | .88 | .76 |
| 076 | DISGUST | 173 | .29 | .22 | 1.05 | 1.03 |
| 077 | SADNESS | 121 | 1.87 | .15 | 1.04 | 1.02 |
| 078 | SADNESS | 159 | .84 | .18 | 1.03 | 1.06 |
| 079 | HAPPINESS | 194 | −1.43 | .42 | .90 | .78 |
| 080 | DISGUST | 126 | 1.75 | .15 | .98 | .95 |
| 081 | DISGUST | 184 | −.34 | .27 | 1.07 | 1.12 |
| 082 | ANGER | 160 | .80 | .19 | 1.02 | 1.05 |
| 083 | FEAR | 178 | .04 | .23 | 1.04 | 1.10 |
| 084 | SADNESS | 144 | 1.29 | .17 | 1.03 | 1.04 |
| 085 | HAPPINESS | 195 | −1.63 | .46 | .89 | .77 |

**Table 5** (*continued*)

| Item | EMOTION | Score | Rasch *Di* | SE | *Infit* | *Outfit* |
|------|---------|-------|------------|-----|---------|----------|
| 086 | ANGER | 170 | .42 | .21 | .97 | .88 |
| 087 | ANGER | 166 | .59 | .20 | 1.02 | 1.05 |
| 088 | DISGUST | 171 | .38 | .21 | 1.09 | 1.19 |
| 089 | SADNESS | 161 | .77 | .19 | 1.00 | 1.01 |
| 090 | HAPPINESS | 198 | −2.59 | .72 | .84 | .19 |
| 091 | DISGUST | 177 | .09 | .23 | 1.02 | .94 |
| 092 | FEAR | 160 | .80 | .19 | .96 | .88 |
| 093 | HAPPINESS | 196 | −1.87 | .51 | .85 | .29 |
| 094 | HAPPINESS | 190 | −.87 | .33 | .96 | 1.19 |
| 095 | SADNESS | 164 | .66 | .19 | 1.04 | .99 |
| 096 | ANGER | 176 | .14 | .23 | .97 | .89 |
| 097 | SADNESS | 115 | 2.01 | .15 | 1.02 | 1.03 |
| 098 | SADNESS | 175 | .20 | .22 | .97 | .82 |
| 099 | DISGUST | 142 | 1.35 | .16 | .98 | .96 |
| 100 | DISGUST | 191 | −.99 | .35 | .91 | .55 |
| 101 | FEAR | 82 | 2.75 | .15 | 1.11 | 1.18 |
| 102 | FEAR | 171 | .38 | .21 | 1.04 | 1.16 |
| 103 | SADNESS | 184 | −.34 | .27 | .92 | .69 |
| 104 | ANGER | 158 | .87 | .18 | .98 | .91 |
| 105 | FEAR | 192 | −1.12 | .37 | .93 | .57 |
| 106 | FEAR | 175 | .20 | .22 | .90 | .70 |
| 107 | HAPPINESS | 195 | −1.63 | .46 | .85 | .29 |
| 108 | DISGUST | 171 | .38 | .21 | .95 | .88 |
| 109 | SADNESS | 192 | −1.12 | .37 | .96 | .94 |
| 110 | HAPPINESS | 192 | −1.12 | .37 | .90 | .47 |
| 111 | ANGER | 145 | 1.26 | .17 | 1.06 | 1.06 |
| 112 | FEAR | 189 | −.77 | .32 | .88 | .54 |
| 113 | HAPPINESS | 194 | −1.43 | .42 | .82 | .38 |
| 114 | SADNESS | 185 | −.41 | .28 | .92 | .69 |
| 115 | DISGUST | 122 | 1.84 | .15 | 1.20 | 1.28 |
| 116 | HAPPINESS | 194 | −1.43 | .42 | .91 | .55 |
| 117 | ANGER | 168 | .51 | .20 | 1.01 | .95 |
| 118 | ANGER | 165 | .63 | .19 | .96 | .90 |
| 119 | FEAR | 185 | −.41 | .28 | .96 | .80 |
| 120 | DISGUST | 186 | −.49 | .29 | .86 | .56 |

tests, some evidence of multidimensionality should be expected when measuring complex constructs, e.g., when measuring math ability, some evidence of multidimensionality is better tolerated than when measuring geometry aptitude (*Linacre, 2013*).

It is relevant to note here that the use of parametric statistical methods takes for granted interval status, even though the nature of many scoring systems is ordinal at best. We have evaluated the interval scaling assumption with the RM, which because of its desirable metric properties can be used to quantify different types of experimental data (*Delgado, 2007*). Some other advantages of the RM, at the practical level, are the ease of interpreting

and communicating results: because both participants and items are located on the same variable, comparisons can be made concerning what items have been passed by what persons (*Prieto et al., 2010*).

As to gender differences, at least three quantitative reviews have shown clear evidence of female superiority in the accuracy of affective judgments; effect sizes are small-to-medium following conventional standards (*Hall, Gunnery & Horgan, 2016*). In our study, the testing of gender differences was carried out after corroborating that item DIF would not be a plausible alternative hypothesis for the results. No statistically significant sex-related differences were found in EV, CES, FES or total EK score. However, the sign of $d$ indicate that female participants scored consistently higher than male ones, $-.11$ (EV), $-0.21$ (CES), $-0.02$ (FES) and $-.17$ (total EK), a result that will be of interest for future meta-analyses. For instance, a recent multi-level meta-analysis by *Thompson & Voyer (2014)* found that the effect size of the difference in emotion perception, a basic emotional aptitude, is $d = -.19$ (if coded as female = 0, male = 1), not far from the $d = -.17$ found on our study for the EK measures. Given such a small effect size, finding statistically significant sex-related differences in EK would require studies with very large samples.

Finally, when EK items from the three tests were conjointly scaled, item difficulty did not statistically differ as a function of the original test (if CES, FES or EV, ordered by ascending average item difficulty) and so they could be used somewhat interchangeably when measuring EK with time restrictions. This is not implying that the three tests are measuring the same processes (in an essentialist way), only that there is one latent variable (EK), all items tap into it, and the level of this EK variable is in a certain moment the focus of measurement interest (*Wu, Tam & Jen, 2017*). Descriptively, the average item difficulty was ordered as expected from psychological distance theories: CES item scenarios were designed as the most concrete ones, while the EV items, words, were the most abstract stimuli. As to emotions, only HAPPINESS items were significantly easier than the remaining ones, corroborating results from previous research in emotion recognition and emotion understanding (*Delgado, 2012*; *Delgado, 2016*; *Russell, 1994*; *Suzuki, Hoshino & Shigemasu, 2006*). From the perception science field, it has been suggested that the "happiness superiority effect" could have evolved due to the fact that happy faces are communicatively less ambiguous than the remaining facial expressions of emotion (*Becker et al., 2011*).

Thus, the three tests are ready to be used as components of a higher-level measurement process (*Newton & Shaw, 2013*). A promising application field is the assessment of EK as a mediator of change in social competence, given that EK is consistently associated with various social and behavioral outcomes in children and teenagers (*Trentacosta & Fine, 2010*) and EK deficits are found in disorders such as alexithymia (*Lumley, Neely & Burger, 2007*).

## ACKNOWLEDGEMENTS

We want to thank the "Fundación Salamanca Ciudad de Cultura y Saberes" for lending us a quiet space in the DA2 Domus Artium in order to get a heterogeneous participants' sample.

### Funding

This work was supported by the Spanish Ministry of Economy and Competitiveness (MINECO) under Grant PSI2014-52369-P. The funders had no role in study design, data collection and analysis, decision to publish, or preparation of the manuscript.

### Grant Disclosures

The following grant information was disclosed by the authors:
Spanish Ministry of Economy and Competitiveness (MINECO): PSI2014-52369-P.

### Competing Interests

The authors declare there are no competing interests.

### Author Contributions

- Ana R. Delgado conceived and designed the experiments, analyzed the data, contributed reagents/materials/analysis tools, wrote the paper, prepared figures and/or tables, item construction; test computerisation; data collection supervision.
- Gerardo Prieto analyzed the data, contributed reagents/materials/analysis tools, prepared figures and/or tables, reviewed drafts of the paper, item construction.
- Debora I. Burin contributed reagents/materials/analysis tools, reviewed drafts of the paper, item construction.

### Human Ethics

The following information was supplied relating to ethical approvals (i.e., approving body and any reference numbers):

The fact that the Spanish Ministry of Economy and Competitiveness (MINECO) was the funder of the research under Grant PSI2014-52369-P implies that the project was revised and approved. Individual privacy and anonymity were protected. Following usual procedures in psychological research, data was aggregated and participants gave informed consent (the computerised test includes a button "I consent" to start the tasks).

### Data Availability

The raw data has been supplied as a Supplementary File.

### Supplemental Information

Supplemental information for this article can be found online at http://dx.doi.org/10.7717/peerj.3755#supplemental-information.

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
