# Peer review of "Constructing three emotion knowledge tests from the invariant measurement approach"

_PeerJ, doi:10.7717/peerj.3755_

## Round 0.1 · original submission · Major Revisions

This is an interesting paper. It was a pleasant reading. The authors used the Rasch analysis which is one of the most widely used psychometric methods to access general latent traits (Andrich, 2010).
After careful reading and considerations from the reviewers comments, my main concern matches some points raised by the second reviewer and it is precisely about the strength of the method generalization which in turn leads to its results validity. However, I truly think that once the authors consider the corrections and clarifications suggested the paper will be suitable to be published.

·

Basic reporting

'no comment'

Experimental design

'no comment'

Validity of the findings

'no comment'

Additional comments

I followed all the reasoning and quite agree with the option for testing using the RM. I am not skilled in the statistics involved in this kind of measurement. My recommendation is based in the quality of the reasoning and ingenuity of the proposed method and therefore does not follow a proper analysis of the data treatment.

Reviewer 2 ·

Basic reporting

In this paper, Delgado et al test 3 emotion knowledge tests in a healthy adult Spanish population (i) Emotion Vocabulary; (ii) Close Emotional Situations and (iii) Far Emotional Situations. They use Rasch analysis to assess the construction and validity of these tests in the context of the Conceptual Act Theory (CAT) model. While the objectives of the paper are interesting, and the Rasch modelling appears to be well conducted, the methodlogical detail provided in the manuscript is currently insufficient. Areas which require further information are detailed below.

While the authors provide reasonable justification for the use of Rasch analysis no hypotheses were provided. Furthermore, it is unclear why the authors first analyse performance on each test separately and then together. The comment that "when EK items from the three tests were conjointly scaled, item difficulty did not statistically differ as a function of the original test and so they could be used somewhat interchangeably when measuring EK with time restrictions" is not well justified. Do the authors suggest that because these tests are not statistically different then they are measuring the same underlying psychological construct? Confirmatory factor analysis is typically employed following Rasch analyses to justify such claims, although this analysis does not appear to have been conducted here.

Experimental design

My major concern is the inadequacy of the description of the tasks employed. The authors should provide more detail about how each of the three tests were constructed. For example, the number of items per test; example stimuli; the amount of time available to respond; whether participants were provided with feedback throughout. How were items scored? A Figure demonstrating example trials for each task would be helpful.

Information about ethics is currently absent. Which ethics committee approved the study? How was consent provided?

How generalisable to the wider population are these findings given that all participants were recruited from an art gallery? Do people with better "emotion knowledge" attend art galleries? What is the potential impact of this convenience sample on the interpretation of the findings?

Validity of the findings

The authors make some comments about the potential generalisation of the findings "Although items were generated in Spanish, the careful selection of the emotion words and the substantive background should facilitate their adaptation to other languages or cultures" [Discussion, para. 1], this is not convincing. The authors should provide information about how these tests could be translated. Would direct translation of the items be sufficient?

·

Basic reporting

Minor unclear terms and sentences, described on the annotated PDF.

Experimental design

No comment

Validity of the findings

No comment

---

## Round 0.2 · accepted · Accept

Thank you for your patience.

·

Basic reporting

No comment.

Experimental design

No comment.

Validity of the findings

No comment.

Additional comments

No comment.